# A New Species of *Hemilauxania* Hennig (Lauxaniidae) from the Lower Eocene Oise Amber—The Oldest Record of Schizophora (Diptera)? [note 1]

**DOI:** 10.3390/insects14110835

**Published:** 2023-10-25

**Authors:** Jindřich Roháček, Christel Hoffeins

**Affiliations:** 1Entomology, Silesian Museum, Nádražní okruh 31, 746 01 Opava, Czech Republic; 2Independent Researcher, Liseistieg 10, D-22149 Hamburg, Germany; chw.hoffeins@googlemail.com

**Keywords:** extinct lauxaniid flies, age of Acalyptratae, Paleogene

## Abstract

**Simple Summary:**

The first finding of a schizophoran fly in Oise amber (France) proved to belong to an extinct genus *Hemilauxania* Hennig, 1965 (Acalyptratae: Lauxaniidae), known previously only from Baltic amber (ca 48–34 Ma). It is described as a new species, *H. parvula*, and because it originates from the lower Eocene (about 53 Ma) it is apparently the oldest known representative of Acalyptratae as well as of Schizophora (Diptera).

**Abstract:**

*Hemilauxania parvula* sp. nov., a new fossil species of the family Lauxaniidae (Diptera: Acalyptratae), is described and illustrated from Oise amber, France (Eocene, lower Ypresian, ca 53 Ma), and its relationship is discussed. Inasmuch as this first finding of a member of Schizophora in Oise amber probably represents the oldest known record of this group of Diptera, the age of Schizophora, based on the known fossil records, is discussed.

## 1. Introduction

The recent Lauxaniidae are a rich and diversified group of Diptera Acalyptratae [1]. However, they are scarce in the fossil record. Only two genera and two species have been described from Baltic amber (Eocene, 48–34 Ma), viz. *Chamaelauxania succini* Hennig, 1965 and *Hemilauxania incurviseta* Hennig, 1965 [2], but there is a series of lauxaniid specimens of the same origin in collections probably also containing unnamed species, see [3,4]. Hong [5] described *Trypaneiodes ellipticus* from the Eocene (Lutetian, 41.2–47.8 Ma) Fushun amber (China), included it in Lauxaniidae, and subsequently transferred it to a new genus, *Eolausanites* Hong, 2001, under the family “Lausaniidae”, see [6] (p. 412). Both these names were subsequently corrected to *Eolauxanites* and Lauxaniidae in an atlas by Hong [7] (p. 206), where photos of the holotype *E. ellipticus* (Hong, 1981) are presented. However, the affiliation of this fossil taxon to Lauxaniidae is not certain. According to Gaimari and Miller [1] (p. 1764), at least 15 undescribed species of Lauxaniidae are known from the early Miocene Dominican amber (Burdigalian, 16–18 Ma), but they all belong to extant New World genera. Also, two compression fossils of Lauxaniidae from the Oligocene are described in the literature, one from Colorado, USA, viz. *Sapromyza veterana* Melander, 1949 [8], the other from British Columbia, Canada, viz. *Lonchaea senescens* Scudder, 1877 (originally placed in Lonchaeidae [9], but see [10,11]). However, their taxonomic status and systematic classification are in need of revision. Consequently, all known fossil Lauxaniidae are from the Paleogene, and none of them are older than 48 Ma.

A new fossil species of Lauxaniidae (Figure 1), described below, belongs to the genus *Hemilauxania* Hennig, 1965, being formerly only known from Baltic amber [2,3]. It was found in a sample of Oise amber (France). The fauna and flora of the amber from Le Quesnoy at the Oise department, Paris Basin, France are relatively well documented. Amber from Le Quesnoy, usually addressed as Oise amber, is of lowermost Eocene age (Sparnacian, lower Ypresian, basal Eocene, 53 Ma). An inventory list of taxa described from Oise amber was given by Brasero et al. [12]. All main arthropod orders known from other Eocene amber deposits, like that from the Baltic Bay of Gdańsk, are present in this list except for scorpions and myriapods. Diptera were represented by only 12 species belonging to six families, but, subsequently, a series of further taxa from several families has been added [13,14,15,16,17,18,19,20,21,22]. Currently, more than 50 species belonging to 14 families of Diptera are known from Oise amber, including nematocerous Bibionidae, Anisopodidae, Mycetophilidae, Cecidomyiidae, Keroplatidae (as Lygistorrhinidae), Chironomidae, Psychodidae and Scatopsidae and brachycerous Athericidae, Rhagionidae, Bombyliidae, Mythicomyiidae, Scenopinidae, Hybotidae, and Dolichopodidae. Consequently, no representative of higher flies (Schizophora) has hitherto been recorded from this amber deposit.

The oldest records of ancient Schizophora (both Acalyptratae and Calyptratae) have recently been reviewed in this journal by us [23]. In that review, we considered as such those from the Eocene Baltic amber (48–34 Ma). However, there is an older fossil species of the group, viz. *Pareuthychaeta eoindica* Grimaldi, 2012, belonging to Ephydroidea [24] and originating from Cambay amber (India) of the Early Eocene (Ypresian, 50–52 Ma) age, see [25,26]. Most recently, the oldest representative of Calyptratae, belonging to Hippoboscidae, viz. *Eornithoica grimaldii* Nel, Garrouste & Engel, 2023, has been described from a compression fossil found in the Green River Formation, Colorado, USA (lower Eocene, Ypresian, 46.2–40.4 Ma), but the authors suggested its age to be around 52 Ma based on radiometric timing of the same fossil deposit [27]. For completeness, it should be mentioned that there are findings of compression fossils of *Platanus* leaves with mines attributed to Agromyzidae from southeastern Montana (USA) dated to the early Paleocene (ca 65 Ma) [28,29]. However, this record, considered by authors the earliest evidence for schizophoran flies, has not been confirmed by finding of fossil adults to support that statement.

The aim of this paper is not only to describe the oldest known species of Lauxaniidae but also to discuss its relationships and palaeohabitat as well as the assumed age of Acalyptratae and Schizophora as a whole.

## 2. Materials and Methods

### 2.1. Material

Origin and repository. The amber sample with the *Hemilauxania* inclusion has been registered in the Hoffeins’ collection under inventory number CCHH #729-11 but currently is deposited in the Palaeontological collection, Muséum national d’Histoire naturelle, Paris, France (MNHN). The inclusion was a donation to the junior author from an amber friend who purchased a large lot of Oise amber with inclusions at the mineral fair at Sainte-Marie-aux-Mines (France) somewhat in the beginning of the 2000s. The very first collector of this amber from Oise is unknown [30].

### 2.2. Preparation of Amber Specimen

The amber with inclusion was cut, ground and polished and subsequently embedded in Polyester resin for conservation purposes following the method described in [31]. However, for a better view and photography, the artificial resin was removed from one side of preparatum (Figure 2A,B) and this side of the amber was cleaned and polished during the examination process (Figure 2C).

### 2.3. Techniques of Investigation

The amber inclusion was examined, drawn and measured using two types of binocular stereoscopic microscopes (Reichert, Olympus). The specimen was photographed by means of a Canon EOS 60D digital camera with macro lens Canon MPE 65 mm 1–5× and, particularly, by a Canon EOS 5D Mark III digital camera with a Nikon CFI Plan 4×/0.10 NA 30 mm WD objective attached to a Canon EF 70–200 mm f/4 L USM zoom lens. During photography by the latter equipment, the specimen was repositioned upwards between each exposure using a Cognisys StackShot Macro Rail, and the final photograph was compiled from multiple layers (35–40) using Helicon Focus Pro 7.0.2. The final images were edited in Adobe Photoshop CS6. Other illustrations were drawn on the basis of macrophotographs in which details were inked based on direct observation at higher magnification using a binocular microscope. Description of the species was prepared following the sequence of morphological structures and characters as developed for fossil acalyptrates by [23,32]. Measurements: Six characteristics were measured—body length (measured from anterior margin of head to end of cercus, thus excluding the antenna), wing length (from wing base to wing tip), wing width (maximum width), index Cs_3_:Cs_4_ (=ratio of length of 3rd costal sector to length of 4th costal sector), index r-m\dm-cu:dm-cu (=ratio of length of section between r-m and dm-cu on discal cell to length of dm-cu) and index r-m\dm-cu:CuA_1_ (=ratio of length of section between r-m and dm-cu on discal cell to length of apical portion of CuA_1_).

### 2.4. Morphological Terminology

The morphological terminology follows that used in [23] for ancient Clusiomitidae, to be in continuation with [32] on fossil Anthomyzidae. Morphological terms of the female abdomen are depicted in Figures 5 and 8. The synonymous morphological terms of adult structures and their abbreviations as used in the most recent manual of Afrotropical Diptera [33] are given in parentheses in the list of abbreviations (used in text and/or figures) below.

A_1_—first branch of anal vein (=CuA+CuP)A_2_—second branch of anal vein (=A_1_)ac—acrostichal (setulae)bm—basal medial cellC—costace—cercusCs_2_, Cs_3_, Cs_4_—2nd, 3rd, 4th costal sectorCuA_1_—cubitus (=M_4_)cup—posterior cubital cell (=cua)cx_1_, cx_2_, cx_3_—fore, mid, hind coxadc—dorsocentral setaedm—discal medial celldm-cu—discal medial-cubital (=dm-m, posterior, tp) cross-veinf_1_, f_2_, f_3_—fore, mid, hind femurfrt—frontal triangleh—humeral cross-veinhu—humeral (=postpronotal) (seta)ia—posterior intra-alarM—media (=M_1_)mspl—mesopleural (=anepisternal) (seta)npl—notopleural (seta)oc—ocellar (seta)ori—lower fronto-orbital (seta)ors—upper fronto-orbital (seta)pa—postalar (seta)ppl—propleural (=proepisternal) (seta)pra—pre-alar (=anterior supra-alar)(seta)prs—presutural (=presutural intra-alar) (seta)pvt—postvertical (=poc, postocellar) (seta)R_1_—1st branch of radiusR_2+3_—2nd branch of radiusR_4+5_—3rd branch of radiusr-m—radial-medial (=anterior, ta) cross-veinS1–S10—abdominal sternasa—supraalar (seta)sc—scutellar (seta)Sc—subcostasctl—scutellumstpl—sternopleural (=katepisternal) (seta)T1–T10—abdominal tergat_1_, t_2_, t_3_—fore, mid, hind tibiavte—outer vertical (seta)vti—inner vertical (seta)

## 3. Results

### 3.1. Systematic Palaeontology

Class Insecta Linnaeus, 1758.Order Diptera Linnaeus, 1758Cyclorrhapha Brauer, 1863Schizophora Becher, 1882Acalyptratae Macquart, 1835Superfamily Lauxanioidea Macquart, 1835Family Lauxaniidae Macquart, 1835Genus *Hemilauxania* Hennig, 1965

### 3.2. Hemilauxania parvula *sp. nov.* (Figure 1, Figure 3, Figure 4, Figure 5, Figure 6, Figure 7 and Figure 8)

LSID urn:lsid:zoobank.org:act:763D790C-F119-4478-841D-EA25DF67AA24

**Type material.** Holotype female, labelled: “Faszination Bernstein, Christel Hoffeins, Hans Werner Hoffeins” (framed on obverse), “729-11, Diptera, Acalyptratae, Lauxaniidae” (handwritten by C. Hoffeins, on reverse), “Oise amber” (handwritten on pale green label) and “Holotypus ♀, *Hemilauxania parvula* sp.n., J. Roháček & C. Hoffeins det. 2023” (red label). A female in good preservation, embedded in a multi-layered piece of amber of light orange colour; inclusion almost complete, tibiae and tarsi of fore legs and tarsus of one mid leg cut-off, head very close to the surface, face, antennae and anterior part of eyes destroyed, apex of the right wing cut-off. [block-shaped/prism with quadrangular base/amber piece ca 7.3 × 5.1 × 3.4 mm, embedded in polyester resin, actual size 11.2 × 9.4 × 6 mm] (Figure 2); syninclusions: fragment of Chironomidae, female. The holotype is deposited in MNHN, inventory No. MNHN.F-A71404.

**Type locality:** France: Paris Basin, Oise department, Le Quesnoy.

**Horizon and age:** Sparnacian, lower Ypresian, basal Eocene, ca 53 Ma.

**Etymology:** The new species is named “parvula” (=small, minor, from Latin, adjective) to refer to its small body size.

**Diagnosis.** The new species differs from the only other representative of the genus *Hemilauxania*, viz. *H. incurviseta* Hennig, 1965, by a distinctly smaller size (2.7 mm vs. 3.5–5 mm), laterally yellow mesonotum and yellow scutellum, frons finely striated between orbits and frontal triangle, pvt convergent but not crossed, oc shorter, postocular setulae in single row, only 1 robust mspl seta, f_1_ with 2 dorsal setae in distal half, f_2_ with 3 anterodorsal setae near the middle, f_3_ with no anterodorsal setae, t_3_ lacking a group of enlarged anterodorsal setae, C with spinulae ending in front of apex of R_2+3_, costal and subcostal cell distinctly wider, and female T7 with lateral parts extended onto ventral side of postabdomen.

**Figure 3 insects-14-00835-f003:**
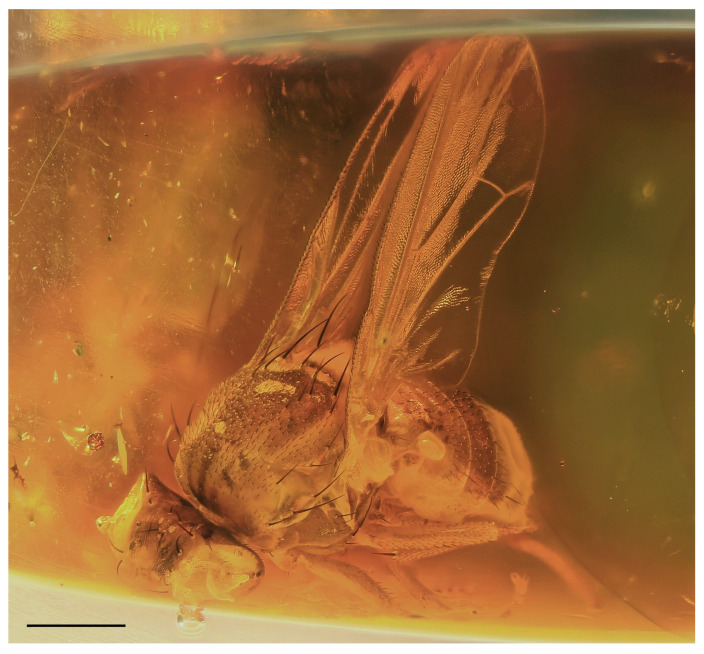
*Hemilauxania parvula* sp. nov., female holotype, dorsally to left dorsolaterally. Scale: 0.5 mm. Photo by J. Roháček.

**Description:** Male unknown. Female. Total body length about 2.7 mm; general colour apparently bicolourous but mostly brown, with only some parts of body ochreous to yellow; legs pale brown to yellow; thorax and abdominal sclerites microtomentose and relatively dull (Figure 1 and Figure 3).

**Figure 4 insects-14-00835-f004:**
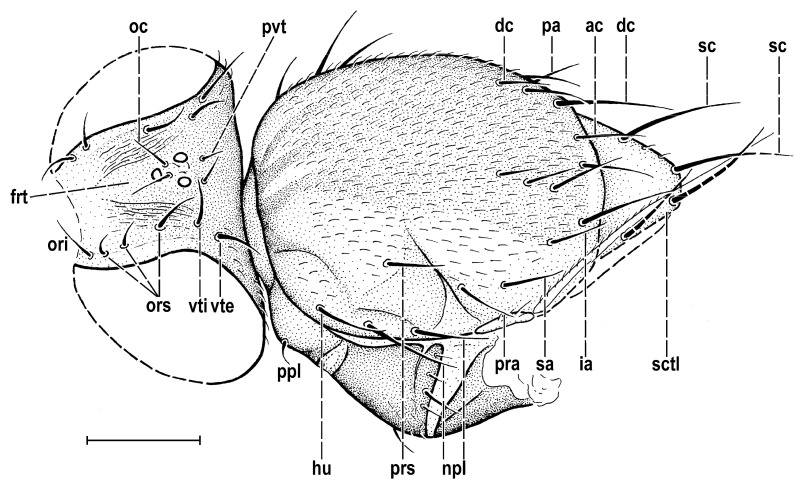
*Hemilauxania parvula* sp. nov., female holotype, head and thorax dorsally, with chaetotaxy. Scale: 0.5 mm. For abbreviations, see Section 2.4.

***Head*** (Figure 3 and Figure 4) slightly higher than long, dorsally about as broad as thorax (not precisely measurable due to damaged eyes); dorsal part of occiput very slightly concave. Head largely brown (almost entire frons, occiput darkest), only some small anterior and ventral parts yellow. Frons about as long as wide and as wide as eye in dorsal view, very slightly tapering anteriorly, largely brown, with only very anterior part of orbit and (probably) anterior margin of frons yellowish ochreous. Orbit delimited by a distinctly striated area separated medially by the frontal triangle. Frontal triangle relatively narrow (but wider than ocellar triangle), brown, with plain and dull surface, with anterior corner poorly visible, probably almost reaching anterior margin of frons. Ocellar triangle not clearly delimited (margined) but distinctly protruding; ocelli medium-sized. Anterior part of head, including frontal lunule, face (praefrons), parafacialia and both antennae damaged, and, therefore, not described. Gena also obscured; postgena pale brown, relatively small, not expanded, with several small setae at posterior margin. Cephalic chaetotaxy (Figure 3 and Figure 4): pvt distinct but shorter than oc, strongly convergent but not crossed; vti, vte and posterior ors relatively strong, subequal (longest cephalic setae); oc somewhat shorter and weaker, inserted between ocelli close to each other; 3 (on right) or 4 (on left) strong fronto-orbital setae, composed of 2 or 3 reclinate ors and 1 inclinate ori (sensu Hennig 1965), the hindmost ors only slightly longer and the foremost ors smaller than others (Figure 4); no microsetulae on frons medially or in front of ors, but 3 microsetae inserted between ocelli on ocellar triangle; postocular setulae numerous, in single long row surrounding posterior eye margin and ranging from vti to postgena. Setosity of damaged anteroventral part of head not visible. Eye relatively large, bare, convex, suboval to subcircular, with longest diameter somewhat oblique and only 1.1–1.2 times as long as shortest diameter. Gena probably low (damaged, not precisely visible); palpus not preserved in the fossil. Mouthparts also damaged, distorted and poorly preserved, yellow. Both antennae heavily damaged, with pedicel and 1st flagellomere (including arista) lost. Only scape partly visible, small, short and yellow, with series of 6 or 7 microsetulae at distal margin dorsally.

**Figure 5 insects-14-00835-f005:**
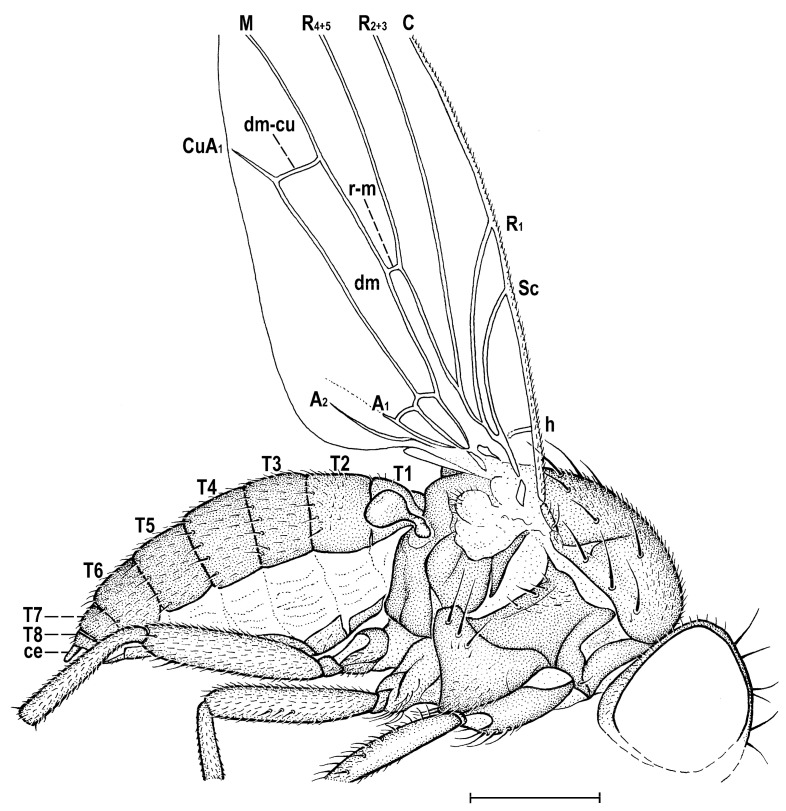
*Hemilauxania parvula* sp. nov., female holotype, laterally, with wing venation. Scale: 0.5 mm. For abbreviations, see Section 2.4.

**Figure 6 insects-14-00835-f006:**
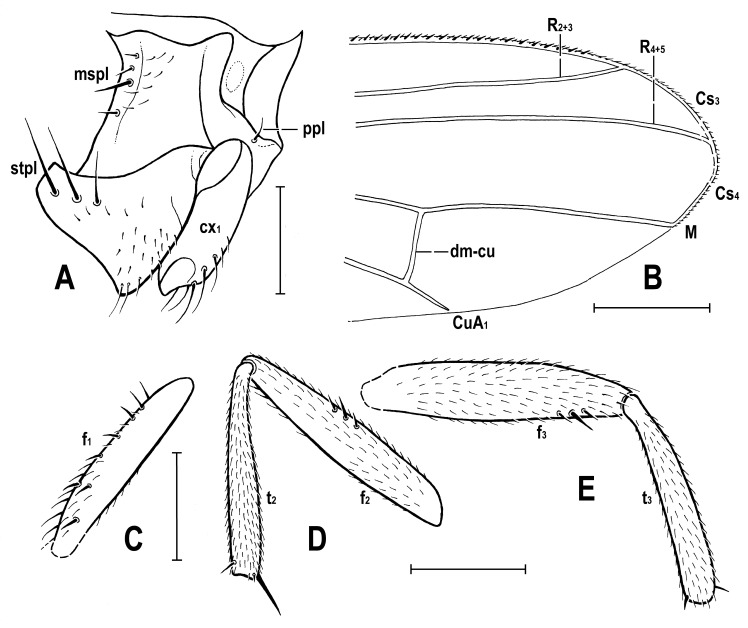
*Hemilauxania parvula* sp. nov., female holotype, laterally, structures of thoracic pleuron, wing and legs: (**A**) anterior part of pleuron, right laterally; (**B**) distal part of left wing; (**C**) right fore femur, dorsally; (**D**) right mid femur and tibia, anteriorly; (**E**) left hind femur and tibia, anteriorly. Scales: 0.3 mm (**A**,**C**–**E**) and 0.5 mm (**B**). For abbreviations, see Section 2.4.

***Thorax*** about as broad as head, bicolourous but largely brown, with only some parts paler, ochreous to yellow (Figure 1 and Figure 3). Mesonotum mostly brown, darker medially and posteriorly, but anteromedially with 3 short dark stripes separated by yellow spots (see Figure 3); also lateral parts of mesonotum and scutellum yellow to pale ochreous, subscutellum brown. Humeral (postpronotal) callus narrowly brown only anteriorly, otherwise yellow; also notopleural area pale brown anteriorly, yellow posteriorly; pleural part of thorax with all sclerites brown (Figure 1). Scutellum rounded trapezoid, wider than long, slightly convex dorsally; subscutellum poorly visible but probably not protruding. Thoracic chaetotaxy (Figure 4 and Figure 5): 1 distinct (as long as posterior npl) hu, plus several microsetae on humeral callus; 2 npl, anterior strong, about 1.5 times as long as posterior; 1 prs (presutural intra-alar), 1 pra (pre-alar = anterior supra-alar) and 1 sa, all well developed but thin; 1 distinct but thin ia (posterior intra-alar); 2 pa, anterior robust, posterior weak; 3 (on right) or 4 (on left) dc, all postsutural and becoming shorter anteriorly; the hindmost dc long and robust (almost as long as sc) and shifted more laterally than others; scutum otherwise covered by uniform and relatively dense microsetae (more than 15 dc microsetae in row in front of anterior dc); ac microsetae also dense, arranged in 8–10 irregular rows in front of suture, in about 6 rows at level of posterior dc; prescutellar ac strong and as long as middle dc and having a pair of ac microsetae between them; 2 strong and long sc (longest thoracic setae), subequal in length; apical sc distinctly crossed, also laterobasal sc more or less inclinate; scutellum with 2 very minute microsetae at margin between laterobasal and apical sc but none on disc (Figure 4); 1 short and fine upcurved ppl (Figure 6A); mesopleuron with only 1 robust but relatively short mspl plus 2 weak setae above and 1 below it at posterior margin (Figure 6A), otherwise with a series of microsetae in posterodorsal half of sclerite; sternopleuron with genus-specific chaetotaxy, i.e., with 3 strong and long stpl at dorsal margin of posterior half and with a number of scattered microsetae on most of surface (including also 2 longer fine setulae on ventral corner).

***Legs*** distinctly paler than thorax (Figure 1 and Figure 7), pale brown to yellow, all relatively robust. Fore and mid legs largely dirty yellow, with some brownish tinge on coxae and femora; hind leg pale brown, with yellowish trochanter and tarsus. Both fore legs with tibia and tarsus missing. cx_1_ with 5 longer fine setae at ventral margin distally; also cx_2_ with a tuft (or short row) of longer fine setae; cx_3_ with only short setulae. f_1_ (Figure 6C) with 2 dorsal setae in distal half, 6 posterodorsal setae in proximal half and a row of 6 or 7 fine posteroventral setae along entire length, all of these setae relatively short; f_2_ (Figure 6D) longer and more robust than f_1_, finely densely setulose but with 3 short but distinct anterodorsal setae about the middle, also 2 subapical posteroventral setae somewhat enlarged; f_3_ (Figure 6E) yet more stout than f_2_, also uniformly densely finely setulose but with 1 robust posteroventral seta in distal sixth, surrounded by 2 thicker but short setae. t_1_ lost, undescribed, but probably with 1 short dorsopreapical seta as known in *H. incurviseta* Hennig, 1965 (see Figure 147 in [2]); t_2_ (Figure 6D) besides dense fine setulae with 1 short dorsopreapical seta and 1 robust and long (longer than width of tibia) ventroapical seta; t_3_ (Figure 6E) also with 1 small dorsopreapical seta and with 1 (yet shorter) ventroapical seta, otherwise uniformly finely setulose. Tarsi simple and short; mid and hind basitarsi (fore tarsi missing) almost as long as other tarsal segments together (see Figure 7); claws relatively small.

***Wing*** (Figure 1, Figure 5 and Figure 6B) proximally widest, distally tapered; veins ochreous; membrane uniformly pale ochreous because densely microtrichose. C reaching to apex of M, besides microsetae provided with series of small spines reaching from beyond apex of humeral cross-vein almost to apex of R_2+3_ (Figure 6B). No costal break. Sc complete, well developed, separate along its entire length, distally ending into C far from apex of R_1_. R_1_ of medium length, bare, ending into C in basal third of wing. Both costal and subcostal cell relatively broad (Figure 5). R_2+3_ long, very slightly sinuate, apically somewhat upcurved to C and ending farther from wing apex than M. R_4+5_ shallowly bent posteriorly, distally subparallel with M (see Figure 6B). Distal part of M also slightly bent posteriorly and ending in C. Cell dm long, gradually slightly widened distally; its upper distal corner rectangular; anterior cross-vein (r-m) situated in the middle of cell dm. Distal part of CuA_1_ slightly (right wing) to distinctly (left wing) shorter than dm-cu cross-vein and almost reaching wing margin. Cells bm and cup closed. A_1_ very shortened, almost stump-like. A_2_ distinctly developed, markedly longer than A_1_ but not reaching wing margin (Figure 5). Anal lobe well developed. Alula small and narrow. Wing measurements: length ca 2.4 mm, width 0.95 mm, Cs_3_:Cs_4_ = ca 1.2, r-m\dm-cu:dm-cu = 2.21–2.44, r-m\dm-cu:CuA_1_ = 1.12–1.32. Haltere (Figure 1 and Figure 5) dirty yellow, knob relatively large, rounded.

**Figure 7 insects-14-00835-f007:**
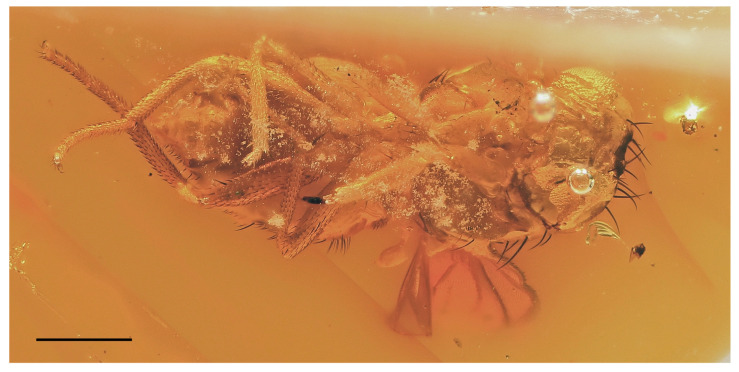
*Hemilauxania parvula* sp. nov., female holotype, ventrally. Scale: 0.5 mm. Photo by J. Roháček.

***Abdomen*** broad, subovoid in ventral view (Figure 7 and Figure 8), posteriorly more tapered. Preabdominal terga short and strongly transverse, all brown and relatively densely but shortly setose, with longest setae at posterior margins (Figure 5). Preabdominal sterna (but only S4 and S5 visible, Figure 8) markedly narrower than associated terga but also short and transverse, sparsely shortly setose. T1 pale brown and narrower than T2, more or less distinctly separate from the latter; T2–T5 darker brown; T2 somewhat widened posteriorly and about as long as T3; T3 widest tergum, T4 of subequal length but slightly narrower; T5 slightly shorter and narrower than T4, distinctly tapered posteriorly, with posterior corners rounded. S1–S3 not visible but probably narrower than S4. S4 and S5 brown, of about the same width, but S4 distinctly longer (Figure 8).

**Figure 8 insects-14-00835-f008:**
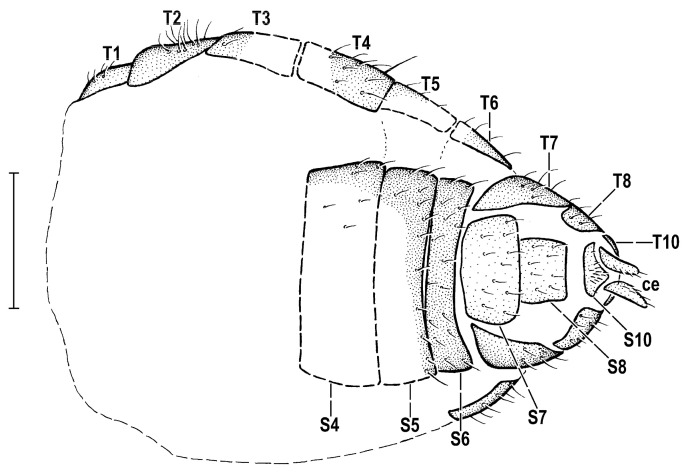
*Hemilauxania parvula* sp. nov., female holotype, abdomen ventrally. Scale: 0.3 mm. For abbreviations see Section 2.4.

***Postabdomen***. Terga only partly visible in lateral (Figure 5) and ventral view (Figure 8). T6–T8 brown. T6 similarly transverse but somewhat shorter and narrower than T5, posteriorly tapered. T7 dorsally distinctly shorter and narrower than T6 but laterally extended to reach on ventral side almost to S7 (see Figure 8), dorsally similarly setose as T6. T8 much smaller than T7, simply transverse, with sides not projecting ventrally. T10 (supra-anal plate) not clearly visible, probably small and pale. Ventral side of postabdomen as in Figure 8. S6 short and very transverse, distinctly shorter than S5 but also brown and with sparse short setae. S7 pale ochreous, narrow (only half width of S6), roughly trapezoidal (anteriorly narrower), laterally rounded. S8 also pale but yet narrower (half width of S7), only slightly wider than long, suboblong. Both S7 and S8 sparsely shortly setose. S10 (subanal plate) small, short transversely subtriangular, with a group of very short dense setulae medially. Cerci (Figure 8) visible as slender, elongately subconical and pale appendages, with a few very short setulae.

**Comments**. *Hemilauxania parvula* sp. nov. is the second species of the formerly monotypic genus *Hemilauxania* Hennig, 1965. However, there are a few more unnamed species of the genus in collections [3,4], all from Baltic amber. The new species, described above, although with anterior part of head and fore legs damaged, undoubtedly belongs to *Hemilauxania*, sharing almost all generic characters presented by Hennig [2]. It seems to be closely allied to *H. incurviseta*, the type species of the genus, but differs from it not only by characters listed above in the Diagnosis, but also by further features, e.g., in having a few microsetae between ocelli, mesonotum anteromedially with distinctive brown and yellow pattern (Figure 3), mesonotal microsetae (acrostichal, dorsocentral and others) distinctly more numerous and denser, scutellum without microsetae on disc, but with 2 marginal ones between sc setae (Figure 4), f_3_ more robust than f_2_, and legs with markedly shorter dorsopreapical setae on tibiae (Figure 6D,E). Hennig [2] did not see vein A_2_ in *H. incurviseta* as being so distinct in *H. parvula* (Figure 1 and Figure 5), but it could be obscured and invisible in the two specimens he examined.

Some variability in setae of head and thorax of the holotype of *H. parvula* should also be commented on here. This specimen has 4 fronto-orbital (3 ors + 1 ori) setae on the left side of head and only 3 (2 ors + 1 ori) on the right side. Similarly, there are 4 postsutural dc macrosetae on the left side of mesonotum and only 3 on the right side (see Figure 4). Comparing this with chaetotaxy found in *H. incurviseta*, see Figures 138–140, 142 and 144 in [2], it is most plausible that 1 ors has been occasionally lost on the right orbital plate of the *H. parvula* holotype (thus, that 3 ors is the normal number) while 1 dc (the foremost) seta seems to be supernumerary on the left side of the mesonotum of that specimen, and normally there should only be 3 dc setae.

## 4. Discussion

The discovery of a new lauxaniid fly in the Oise amber and its affiliation to the genus *Hemilauxania* Hennig lead us to discuss several topics. The general rarity of Lauxaniidae in the fossil record (for review of a few published findings, see Introduction) is reflected in very insufficient knowledge of the evolution of this currently very diverse family of Diptera Acalyptratae. If we ignore the more or less dubious published data (see [5,6,7,8,9]), the only reliable ancient taxa of Lauxaniidae are those based on inclusions in Baltic amber. Hennig [2,34] investigated only five specimens, two males and one female of *Chamaelauxania succini* Hennig, 1965 and two females of *Hemilauxania incurviseta*. In the amber collection of Hoffeins (Hamburg) 28 specimens of Lauxaniidae are actually housed in a total of more than 1100 acalyptrate fossil flies, of which seven are placed in *Chamaelauxania* and 14 in *Hemilauxania*, whereas three cannot be identified to species level because of poor or incomplete preservation, and four may represent an undescribed taxon [4].

### 4.1. Relationships and Classification of Hemilauxania

According to [2], *Hemilauxania* differs from all recent members of Lauxaniidae in having 4 fronto-orbital (including 1 inclinate ori) setae, while only 2 (more rarely 1) ors, sometimes including 1 ori, are present in extant taxa of the family [1,10]. Hennig [35] (p. 33), however, considered the more numerous fronto-orbital setae to belong to the ground plan of Lauxanioidea because 3 (albeit small to hair-like) fronto-orbitals also occur in the closely related family Celyphidae. Although he [35] obviously incorrectly homologized fronto-orbital setae in Celyphidae, see [1], their reduced number (to 2 or 1) can be considered a synapomorphy of recent Lauxaniidae plus †*Chamaelauxania* Hennig, 1965.

The presence of 3 strong sternopleural setae is another distinctive difference against all recent Lauxaniidae (which possess only 1 or 2 stpl) and this character seems to be shared with the other Baltic amber fossil genus, *Chamaelauxania*, having these setae yet more numerous (5 strong stpl), see Figure 131 in [2] (p. 107). It is peculiar that Hennig [2,35] has not discussed this important feature as diagnostic for both his fossil genera; instead, he stressed the presence of 1 pre-alar (=anterior supra-alar) seta in *Hemilauxania* as a unique character within all known Lauxaniidae. Similarly, the presence of a well-developed vein A_2_ was not recorded by him (in both *Chamaelauxania* and *Hemilauxania*), although this vein occurs also on wings of recent Lauxaniidae, cf. [36,37]. It should be remarked that the presence of A_2_ in (many) extant genera of Lauxaniidae has not been listed (neglected or overlooked?) in most Diptera manuals, see [1,10,36].

Although undoubtedly monophyletic, the suprageneric classification of Lauxaniidae “is in its infancy” [1]. As stated already by Hennig [2], *Hemilauxania* and *Chamaelauxania* have no distinct relatives among recent genera of Lauxaniidae and cannot be affiliated to any of the currently recognized subfamilies, viz. Eurychoromyiinae, Homoneurinae or Lauxaniinae [1,36]. Judging from ancestral cephalic chaetotaxy with 3 or 4 fronto-orbital setae (with distinct inclinate ori) and cup cell with a slightly or not concave distal margin (=cross-vein CuA_2_) (see also [2]), *Hemilauxania* should be treated as an extinct lineage of the family, branching very basally from the common stem of Lauxaniidae. The systematic position of *Chamaelauxania* remains unclear; it could either be related to *Hemilauxania* or represents another separate lineage.

### 4.2. Palaeohabitat and Palaeobiology of Hemilauxania

The family Lauxaniidae has an almost cosmopolitan distribution, with most of the species occurring in tropical regions of Africa, Asia and the Americas, and species diversity declines strongly towards the more temperate regions. Most species inhabit humid forests, where the adults are usually collected sitting on leaves of the understory. Larvae are saprophagous and mainly develop in decaying leaves, whether leaf litter in forest or in bird (and mammal) nests [10,37,38,39].

The Eocene Baltic amber and the contemporaneous Rovno amber from the Ukraine were formed in a warm-temperate (“subtropical summerwet”) palaeoenvironment from resin produced mainly by conifers in the so-called “Baltic amber forest”. Alekseev and Alekseev [40] described this palaeohabitat as a thermophilic, humid-mixed forest resembling the recent subtropical forests in eastern and southeastern Asia. However, other authors found that various forest types evolved in this area [41,42,43]. The vegetation obviously had first (in the early Eocene) a tropical character that gradually changed to subtropical and later to warm-temperate due subsequent cooling. On the other hand, the older and more southerly situated (at 42–47° N latitude) Oise amber forest was growing close to the early Eocene warming maximum (ca 49 Ma) and, therefore, plausibly had a tropical character; moreover, the Oise amber had an angiosperm origin, because it was formed from resin produced by the ancient tree *Aulacoxylon sparnacense* Combes, 1907 (Combretaceae or Caesalpiniaceae or Fabaceae), see [44,45]. Nel & Brasero [44] wrote about this palaeohabitat: “In Oise, the dominance of an arborescent amber-producing tree and the presence of freshwater suggest a semi-deciduous forest, with a mosaic of gallery-forest mixed with dryer plant communities, in a deltaic ‘paratropical’ region”.

However, the palaeohabitats with riparian forests and deciduous trees under high humidity conditions, most suitable for development of Lauxaniidae, were documented both for the Oise [12,44] and the Baltic amber [43]. Consequently, the presence of *Hemilauxania* species in amber from both these deposits is not so surprising, particularly when some (or all) Baltic amber samples with *Hemilauxania* inclusions could be older than currently dated (see below). Nevertheless, the occurrence of *Hemilauxania* species in Ukrainian Rovno amber can be predicted with high probability.

### 4.3. Age of Hemilauxania and Other Oldest Fossils in Schizophora

The new *Hemilauxania* species originates from a stratum of the Oise or Le Quesnoy amber being of the Sparnacian, lower Ypresian, basal Eocene age (ca 53 Ma). Factually, it represents the oldest known species not only of Acalyptratae but also of the whole Schizophora. The previously oldest records of schizophoran flies were also lower Ypresian *Pareuthychaeta eoindica* Grimaldi, 2012 (Ephydroidea) from Cambay amber (India) dated 50–52 Ma [24] and *Eornithoica grimaldii* Nel, Garrouste & Engel, 2023 (Calyptratae: Hippoboscidae), a compression fossil from the Green River Formation (Colorado, USA) dated about 52 Ma [27].

In the geological timescale, the Ypresian is the oldest period or lowest stratigraphic stage of the Eocene. It spans the time between 56 and 47.8 Ma, and is followed by the Eocene Lutetian Age. Apart from the above ambers and shales, also the Fur Formation in Denmark and the Messel shales in Germany are of the same Ypresian age.

However, *Hemilauxania incurviseta* was found in the Baltic amber [2], whose deposit and age must also be considered. Baltic amber was deposited in marine sediments in the Eocene epoch. Amber-bearing layers were sedimented in the Lower Blue Earth (Ypresian), the Wild Earth and Blue Earth (Lutetian) and the Green Wall (Bartonian) horizon. The Lutetian Blue Earth is the main amber horizon, but the Lower Blue Earth and the Wild Earth also contain scattered amber [46]. The Lower Blue Earth is an amber-bearing layer of Ypresian age, 54.5–49 Ma.

The age of these Baltic sediments covers a range of roughly 15 Ma. Amber found in these sediments was redeposited in Eocene marine sediments as well as Neogene fluvial and glacigenic deposits [47]; consequently, it is never found in its original stratigraphic position. Moreover, the stratigraphic age does not correlate with the “real” age of the resin that was transformed into amber. If an individual amber nugget is obtained, nothing can be stated about which stratum it may have come from, if Ypresian, Lutetian or Bartonian sediments. We can only state the age of the sediments ranging between 54.5 Ma to 37 Ma. Amber as an exudate of Eocene conifers must have been produced somewhat before it was sedimented; this means it is in fact older than the sediment itself. Consequently, although the main amber-bearing layer from the Baltic area is of Lutetian age, we cannot exclude a possibility that *H. incurviseta* and/or other hitherto undescribed *Hemilauxania* specimens of the same origin also are of Ypresian age, as is *H. parvula* from Oise amber.

The origin of Schizophora (and also Acalyptratae) is estimated by Grimaldi and Engel [48] to have occurred immediately after the K-T (K-Pg) boundary, being associated with the Cretaceous–Paleogene extinction event dated ca 66 Ma [49]. Based on molecular-based time-calibrated phylogeny, Wiegmann et al. [50] reached almost the same conclusion, dating the Schizophora split off at 65 Ma. Considering the age of the oldest known fossils of schizophoran flies (ca 53 Ma, see above), we can agree with the latter dating of the origin of Schizophora because 1 million years after the Chicxulub bolide impact causing the massive K-Pg extinction of fauna, the atmospheric conditions had already been stabilized [49] to enable development of a new group of Diptera.

## Figures and Tables

**Figure 1 insects-14-00835-f001:**
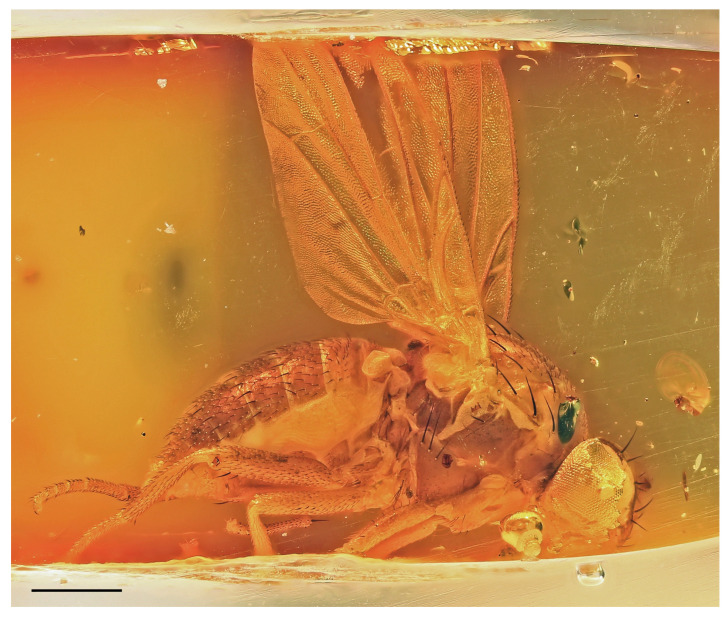
*Hemilauxania parvula* sp. nov., female holotype, right laterally. Scale: 0.5 mm. Photo by J. Roháček.

**Figure 2 insects-14-00835-f002:**
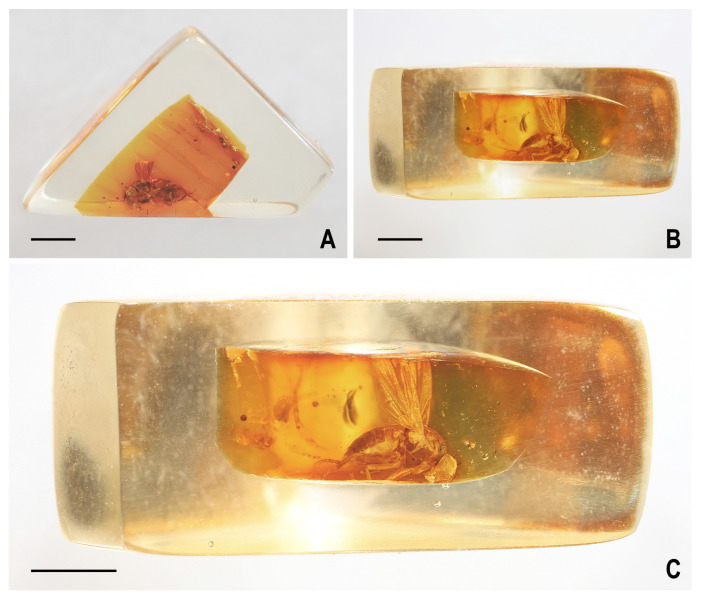
*Hemilauxania parvula* sp. nov., female holotype, preparatum: (**A**) whole amber sample (preparatum in polyester resin), in situ; (**B**) ditto, from largest side; (**C**), ditto, close-up of amber sample. Scales: 0.2 mm. Photos by J. Roháček.

## Data Availability

The data presented in this study are available in this article.

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
