# Peer review of "A New Species of Hemilauxania Hennig (Lauxaniidae) from the Lower Eocene Oise Amber—The Oldest Record of Schizophora (Diptera)?"

_insects, 2023, doi:10.3390/insects14110835_

Round 1

Reviewer 1 Report

There are very few recorded fossil species of the family Lauxaniidae, and the article comprehensively introduces the recorded fossil species of the family. This is a very valuable work describing a new fossil species, which is the oldest recorded species of Diptera. Here are some of my suggestions.

1. The new species described in this article is the second species of the genus Hemilauxania, which is significantly different from the first. Revising the generic characters based on these two species would be a meaningful work.

2. There are some inconsistencies in the Diagnosis, such as line 213 'Head dorsally about as broad as thorax (not precisely measurable due to damaged eyes)' and line 249 'Thorax hard wire than head'.

3. Line 399 mentions that there are currently two subfamilies in the Lauxaniidae, which should be three subfamilies, with an additional subfamily of Eurychoromyiinae added.

Author Response

Dear reviewer,

I have corrected the MS with respect to your suggestions 2 and 3.

However, the generic characters of Hemilauxania (your suggestion no. 1) have not been updated simply because there are a few more undescribed species (from Baltic amber) which should also be taken in consideration. Thus, a new generic diagnosis would be premature. Moreover, with addition of H. parvula the original generic diagnosis of Hemilauxania by Hennig (1965) is unnecessary to change.

As regards the English: the text of the MS (including all its modification following reviews) has been carefully revised lingustically by Peter J. Chandler, the English dipterist and editor of the periodical Dipterists Digest.

Yours, Jindrich Rohacek

Reviewer 2 Report

Overall this is an excellent paper describing a new fossil species of lauxaniid, and is a particularly exciting discovery in my opinion. The description is very well constructed, detailed, and complete, which is very important when dealing with such a rarity. Of course there are some characteristics of this species (and Hemixantha incurviseta) that would leave one wondering if it is really a lauxaniid (the most obvious example being the fronto-orbital setae), but there is no reason to expect such an early member of the lineage retained such characteristics. I agree this is a lauxaniid, and I also agree that this is Hemixantha. Overall this manuscript was a pleasure to read!

Line 62: “the oldest records” – this is a very good discussion of these earliest records demonstrating that this is the oldest so far discovered, but there are some papers discussing potential agromyzid leaf mines that are older. Maybe they can be discounted as real "records", because they may not be agromyzids, but these papers should at least be mentioned I think. Here are a couple of the papers - I think there are more that are cited therein:

https://www.jstor.org/stable/40802060

https://journals.plos.org/plosone/article?id=10.1371/journal.pone.0103542

Lines 374-375: If there are 7 species of Chamaelauxania and 14 of Hemilauxania in the Hoffeins collection, why was this species alone chosen for description?

Line 382-383: “...because 3 (albeit small to hair-like) fronto-orbitalis also occur in the closely related family Celyphidae…”. Although Hennig suggested that the lauxanioid ground-plan is to have 3 fronto-orbital setae (which is reasonable given the evidence presented by Hemixantha, although Chamaelauxania has only 2 (recognizing the supernumerary seta on the right side of the holotype)), his reasoning is off. Celyphidae do not have 3 fronto-orbitals, or at least don’t have setae that can positively be asserted as fronto-orbitals in my opinion. Additionally, Chamaemyiidae are also restricted to 2 fronto-orbital setae, which are reduced, fewer, or absent in Leucopinae.

Line 408: I would suggest that most species occur in the tropical regions of the world, rather than excluding the Afrotropics. Don’t mistake the lack of work on the Afrotropical lauxaniid fauns as a proxy for a lack of diversity! Maybe not equal to the Neotropics or tropical Asia, but extremely diverse.

Line 410: At least some taxa (e.g., Eurychoromyiinae) are known to be in the canopy, and I would suggest that “usually found” mistates the situation since we really do not know what is usual. We can say we usually collect them sitting on leaves of the understory, but that doesn’t mean that’s where they usually are.

Line 410-411: “and develop in rooting wood and, particularly, leaf-detritus” – first, I think you mean “rotting wood”, but that said, this is an incorrect characterization of their biology. They are very rarely associated with wood (a few under rotting bark), but they are heavily associated with decaying leaves, whether leaf litter in the forest or in birds nests. I know you aren’t doing a thorough review of their biology, but I would never say lauxaniids develop in rotting wood. Besides this, I think the discussion of palaeohabitat and palaeobiology is quite good.

Author Response

Dear reviewer,

We have accepted all yout suggestions except for that for lines 374-375 and corrected/supplemented the MS accordingly. 

There are not 7 species of Chamaelauxania and 14 of Hemilauxania but 7 and 14 specimens of these  genera in the Hoffeins collection. Most of them belong to C. succini and H. incurviseta although there could be one or two unnamed Hemilauxania species among them which need a more thourough study in future. The species from Oise amber has been described separately to highlight the importance of this (oldest record of Schizophora) record.

Very best regards, yours

Jindrich Rohacek